# Changes in the Choroidal Thickness of Children Wearing MiSight to Control Myopia

**DOI:** 10.3390/jcm11133833

**Published:** 2022-07-01

**Authors:** Francisco Luis Prieto-Garrido, Cesar Villa-Collar, Jose Luis Hernandez-Verdejo, Cristina Alvarez-Peregrina, Alicia Ruiz-Pomeda

**Affiliations:** 1Hospital Universitario del Henares, Fundación para la Investigación e Innovación Biomédica, FIIB HUIS HHEN, 28822 Madrid, Spain; franciscoluis.prieto@opt.ucm.es; 2Faculty of Optics and Optometry, Complutense University of Madrid, 28037 Madrid, Spain; jlhernan@ucm.es; 3Faculty of Biomedical and Health Sciences, Universidad Europea de Madrid, 28670 Madrid, Spain; villacollarc@gmail.com; 4Ophthalmology Department, Hospital Universitario de Móstoles, 28935 Madrid, Spain; aliciaruizpomeda@hotmail.com

**Keywords:** MiSight, myopia, choroidal thickness, contact lenses

## Abstract

Background: Due to the importance of choroidal thickness in the development of myopia, this study examined the effect of MiSight contact lenses (CLs) on the choroidal thickness of myopic children and the differences between responders and non-responders to the treatment with these CLs. Methods: A total of 41 myopic children were fitted with MiSight CLs and 33 with single-vision spectacles. They were followed up for two years. Subfoveal choroidal thickness and choroidal thickness 1 and 3 mm temporal and nasal to the fovea were measured by OCT at baseline and one and two years after the treatment. Differences in all the choroidal thickness parameters were assessed in each group over time. Patients from the MiSight group were classified based on a specific range of changes in axial length at the end of the second year of treatment as “responders” (AL change < 0.22 mm/per year) and “non-responders”, and the choroidal thickness of both groups was analyzed. Results: The subfoveal choroidal thickness of the MiSight and single-vision spectacle groups did not show any changes over time. Wearing MiSight CLs induced relative choroidal thickening in the responder group in the first year of treatment. Conclusion: Choroidal thickness might work as a predictor of the effectiveness of MiSight in myopia treatment.

## 1. Introduction

Experimental models of myopia and visual regulation of ocular growth have been proven in many species from primates to invertebrates. These animal studies have shown that the eye can develop myopia in response to visual form deprivation and hyperopic defocus by modifying axial length and the choroid [1]. In this process, it has been revealed that biochemical signaling cascades transduce defocused visual stimuli into cellular and biochemical changes in the retina, the retinal pigment epithelium (RPE), and the choroid [2]. The primary role of the choroid is to provide oxygen and nutrients for the external retina and to contribute to the regulation of ocular temperature and intraocular pressure [3], but the participation of the choroid in the regulation of ocular growth has also been postulated in studies showing that refractive errors correlate with alterations in choroidal thickness (ChT) [4]. Animal studies have demonstrated thinning of the choroid during myopia development [5] and thickening in hyperopic animal eyes [6]. Human studies have also shown an association between ChT changes and axial eye growth, observing choroidal thinning in children undergoing faster axial eye growth [7]. In this sense, the choroid is considered to play a key role in visually guided eye growth due to ChT changes in response to retinal defocus [3]. It has been suggested that choroidal thickening is a precursor for reduced eye growth and slowed myopia progression [8]. Although the mechanism underlying the modulation of ChT still remains unknown, these changes in ChT are thought to be produced by a redistribution of fluids in the lacunar space. The hypotheses being considered are as follows: an increased thickness achieved by an increase in the amount of highly charged proteoglycans in the choroidal extracellular matrix, thereby causing swelling of the choroid [5]; an increase in the size or number of choriocapillaris fenestrations [9,10]; entry of drainage fluid from the anterior chamber to the choroid [5]; and modification of the transport of fluid through the retinal pigment epithelium [11,12]. Furthermore, human studies have shown that the human eye can also detect the sign of imposed optical defocus and undergo compensatory changes in axial length and choroidal thickness [6,7,13]. Wang et al. [6] and Chakraborty et al. [13] showed that short-term exposure to myopic defocus induced relative choroidal thickening while hyperopic defocus led to choroidal thinning in children. Further, Read et al. demonstrated that children with faster axial elongation had less thickening of the subfoveal choroidal thickness (SFChT) [7].

These results provide a solid scientific basis for optical treatment strategies for controlling abnormal eye growth to reduce the progression of myopia in children. Orthokeratology (OK) [14,15,16], atropine, soft peripheral-defocus-modifying contact lenses (CLs) [17,18,19,20], and multi-segment spectacle lenses [21,22] have been used to control myopia progression in children. Several studies have shown the effect of OK [23,24,25,26,27] on choroidal thickness. The effect of atropine eye drops on choroidal thickness [28,29] and the additive effect of atropine eye drops and short-term retinal defocus with spectacles on ChT in myopic children have also been studied. However, there are no studies that have evaluated the effect of soft peripheral-defocus-modifying CLs on choroid thickness in myopic children [8]. There is only one previous study [30] that has assessed the impact of two multifocal CLs on ChT in adults.

The MiSight CL is a soft (hydrophilic) single-use, daily disposable contact lens. It contains a concentric design with a large central correction zone surrounded by a series of treatment and correction concentric zones of alternating distant and near powers, which together produce two focal planes. The optical power of the correction zones corrects the refractive error while the treatment zones produce 2.00 diopters (D) of simultaneous myopic retinal defocus during both distance and near viewing, maintaining good visual acuity [20].

This study aimed to determine the influence of wearing MiSight CLs on the choroidal thickness, and to know the differences in choroidal thickness between responders and non-responders to the myopia control treatment with MiSight.

## 2. Materials and Methods

### 2.1. Subjects

This study is a part of the previous study MASS [20] that was designed to assess the efficacy of MiSight^®^ CLs versus distance single-vision (SV) spectacles in myopic children. Eligible subjects were randomized into either the study group (MiSight CLs) or the control group (single-vision spectacles (SV), Shamir, Spain). The subjects of the study group, MiSight CL users, were also divided into two subgroups: responders (R) and non-responders (NR) in the second year of treatment [31].

### 2.2. Ethical Aspects

The study was conducted following the tenets of the Declaration of Helsinki and was approved by the CEI-R (Regional Research Ethics Committee of the Community of Madrid, Spain), with protocol number P2013/05 in November of 2016. After receiving an explanation of the nature and possible consequences of the study, all parents provided signed permission for their children to participate, and participants provided written consent.

### 2.3. Visits and Protocol

Baseline and 12- and 24-month visits included case history, cycloplegic autorefraction measured with an autorefractor (Topcon RM 8000, Tokyo, Japan), best-corrected distance and near visual acuity, biomicroscopy, axial length, and corneal power measured using an IOLMaster (Carl Zeiss Jena GmbH, Jena, Germany), and ChT measurements with Heidelberg optical coherence tomography. The goal of the present study was to investigate changes in SFChT in the center of the fovea and 0.5 and 1.5 mm nasal and temporal to the fovea in children wearing MiSight CLs compared with children corrected with SV spectacles, and to additionally compare changes in ChT between the responder and non-responder groups. To obtain these data, Spectralis spectral-domain optical coherence tomography (SD-OCT) (Heidelberg Engineering Inc., Heidelberg, Germany) was used to measure SFChT. It provides cross-sectional images with an axial resolution of 3.9 microns for chorioretinal images [26]. To obtain adequate images of the choroid, the enhanced depth imaging (EDI-OCT) mode was used. The EDI mode focuses the instrument closer to the posterior eye than the standard imaging mode which allows improved visualization of the choroidoscleral junction and enables valid measurement of ChT at various locations within the macular region [32].

For this, the instrument’s automatic real-time eye tracking function was utilized, while the method for visualizing the choroid included enhanced depth imaging with frame averaging (100 B-scans averaged). This method performs an average of 100 B-scans to increase the signal-to-noise ratio and mitigate the scattering of light that occurs when the laser passes through the different tissues, showing high efficiency and repeatability in normal eyes [32,33]. The scanners obtained are processed using software that eliminates noise and improves resolution. Three consecutive images were taken from each eye. The protocol of a horizontal line centered on the fovea was used since it allowed us to obtain the maximum definition in the shortest time, minimizing fixation errors. These measurements were taken at the following points: subfoveal, and 0.5 (1 mm in diameter) and 1.5 (3 mm in diameter) nasal and temporal to the fovea. Therefore, to ensure that all measurements were taken at the same points, the caliper-based software provided by the device was used to locate the center of the fovea and 1 and 3 mm nasal and temporal to the fovea. Additionally, follow-up mode was used to ensure the scans were captured from the same location during multiple visits. Figure 1, Figure 2, Figure 3 and Figure 4 illustrate the scanning protocol used, with an example OCT image from a representative subject. All the ChT measurements were administered consistently by the same optometrist to avoid variability between examiners. The images obtained were reviewed, discarding those of poorer quality. SFChT is influenced by factors such as age, ethnicity, gender, axial length, IOP [3], and systolic blood pressure. The choroid itself undergoes variations in its thickness throughout the day, presenting choroidal thinning in the morning and thickening in the afternoon following the circadian rhythm, with variations of up to 40 microns [14,15]. For this reason, all the choroidal measurements were performed in the same time slot (16:00 to 18:00 h). All the images were taken by the same observer and in the same time slot (between 4 pm and 6 pm) to avoid physiological fluctuations in ChT due to the diurnal choroidal rhythm.

SFChT was defined as the distance between the outer border of the hyperreflective RPE and the inner edge of the hyporeflective suprachoroidal space, which was measured by drawing a line vertical to a line tangential to the foveal contour [34]. High-quality images were taken subfoveal and 0.5 (1 mm diameter) and 1.5 (3 mm diameter) nasal and temporal to the fovea of the children in the MiSight group and the control group at baseline and in the first and second years of follow-up.

OCT Spectralis incorporates automatic segmentation software for 10 retinal layers, including total retinal thickness, nerve fiber layer, ganglion cell layer, inner plexiform and nuclear layer, outer plexiform and nuclear layer, retinal pigment epithelium, and retinal layer. This software allows the user to display all layers or select one or more layers. Once the highest-quality tomography image was selected, the researcher maintained the lines corresponding to the retinal pigment epithelium and the internal limiting membrane. This last line was used in order to be moved point by point and manually define the sclero-choroidal interface. Subsequently, the gauge provided by the instrument was used to perform the measurements. All these steps are graphically detailed in Figure 1, Figure 2, Figure 3 and Figure 4. Figure 1 shows a high-quality tomography image from a subject, with the lines corresponding to the retinal pigment epithelium (bottom red line) and the internal limiting membrane (top red line) marked in red by the automatic segmentation software.

Figure 2 shows when the internal limiting membrane was used to manually mark the division line between the choroid and sclera, moving that line point by point. Figure 3 shows the result of the movement of the line to define the sclero-choroidal interface.

Finally, Figure 4 shows the result with the retinal pigment epithelium layer, which was marked automatically by the software, and the division line between the choroid and sclera, which was marked manually. Figure 4 also shows how the gauge provided by the instrument was used to perform the measurements at the points defined in the study protocol. Therefore, to ensure that all measurements were taken at the same points, the measuring bar provided by the device was used to locate the center of the fovea and 1 and 3 mm nasal and temporal to the fovea. All the OCTs were conducted and analyzed by the same observer to ensure that they were taken from the same point as the first measurement.

### 2.4. Statistical Analysis

For the statistical analysis, data of the study group’s children (MiSight CLs) and the control group (SV) were included in the analysis of SFChT. Data for the dominant eye only were used to avoid the confounding effect of using non-independent data from both eyes. Statistical analysis of the data was performed using the SPSS statistical software package for Windows (IBM C., Armonk, NY, USA). A descriptive analysis was performed that included the mean value and the standard deviation.

The level of statistical significance was set as 5%. The normally distributed differences between the study group and control group were compared with one-way ANOVA. The sample size was determined by the subjects that completed the clinical trial carried out under ClinicalTrials.gov Identifier: NCT01917110.

To facilitate the analysis, the change in ChT was registered in the fovea and 1 and 3 mm diameter nasal and temporal to the fovea between the first year of follow-up and baseline, and in the fovea and 1 and 3 mm diameter nasal and temporal to fovea between the second year and the first year of follow-up. Positive numeric values indicate an increase in ChT, whereas negative values indicate a decrease. All these measurements were taken in the study group (MiSight CLs) and the control group (SV spectacles), and in the responders and non-responders to the treatment in the MiSight group.

The significance of the increments was studied with two paired tests (parametric and non-parametric): a paired Student *t*-test and the Wilcoxon signed-rank test. These tests were used to compare the same variable in two different periods. Finally, to conduct the comparison between groups, an unpaired Student *t*-test was used.

## 3. Results

A total of 74 participants from the MiSight Assessment Study Spain (MASS) completed the study: n = 41 in the MiSight group and n = 33 in the SV group. Table 1 shows cycloplegic autorefraction, axial length, mean keratometry, SFChT, and 0.5 and 1.5 mm nasal and temporal at baseline, the 12-month visit, and the 24-month visit for the MiSight and SV groups.

To facilitate the analysis of the change in choroidal thickness, the change between the first year of follow-up and baseline was recorded, as well as the change between the second and first years of follow-up, for the MiSight group and the control group (SV).

Table 2 shows the changes in ChT in the fovea and 0 and 1.5 mm nasal and temporal to the fovea between the first year of follow-up and baseline. It also shows the change in ChT in the fovea and 0.5 and 1.5 mm nasal and temporal to the fovea between the second year and the first year of follow-up for the MiSight group and the SV spectacle group.

This table shows that there were no statistically significant differences in the variation in ChT between both groups at any of the distances evaluated, despite finding an increase in the subfoveal and nasal thicknesses at 1 mm diameter in the group of patients treated with MiSight compared to thinning in the control group. Such increases in ChT did not occur in the second year of treatment.

Patients from the MiSight group were also classified based on a specific range of changes in axial length at the end of the second year of treatment as “responders” (R) (AL change < 0.22 mm/per year) and “non-responders” (NR) (AL change ≥ 0.22 mm/per year), and ChT of both groups was analyzed. A total of 41 patients treated with MiSight CLs were included in the study as responders (n = 16) and non-responders (n = 25). Responders had a mean myopic progression of −0.23 D and an axial elongation of 0.13 mm over the two years of treatment, and non-responders had a mean myopic progression of −0.6 D and an axial elongation of 0.38 mm over the two years of treatment [20].

Table 3 shows the changes in ChT in the fovea and 1 and 3 mm diameter to the fovea between the first year of follow-up and baseline. It also shows the changes in ChT in the fovea and 1 and 3 mm diameter to the fovea between the second year and the first year of follow-up for the responder and non-responder groups.

The responders to the treatment with MiSight experienced a significant increase in subfoveal ChT, 1 mm temporal and nasal to the fovea, and 3 mm temporal to the fovea in the first year of follow-up. In contrast, in the second year of follow-up, there were no significant changes in ChT for either group.

## 4. Discussion

This is the first study that compares ChT between MiSight wearers and a control group. It is also the first study comparing ChT between responders and non-responders to myopia control treatment with MiSight contact lenses. Therefore, the results of this study will help to gain a better understanding of the influence of this type of lens on the choroidal thickness.

Our results show that there were no differences in the change in ChT between the MiSight group and the control group wearing SV ophthalmic lenses. Considering this, we could say that the MiSight peripheral defocus contact lens does not cause a significant change in ChT when comparing a group of patients treated with this type of lens versus a control group treated with SV ophthalmic lenses. These results agree with those obtained by Breher et al., who studied the influence of multifocal CLs on ChT after short-term wear in adults. They concluded that changes in ChT might not be the main contributor to the protective effect of multifocal CLs in myopia control [30].

Considering the study group, the responder subgroup experienced an increase in the subfoveal choroidal thickness, 1 mm temporal and nasal to the fovea, and 3 mm temporal to the fovea compared to non-responders in the first year of treatment. These results could be the starting point of new research into the use of ChT as a predictor of the effectiveness of myopia control treatment with MiSight. Similar results were found for OK in the study carried out by Li et al., where they suggested that ChT could be a predictor of the effectiveness of OK [25].

The arrival of OCT as a fast and non-invasive method for obtaining cross-sections of the retinal and ChT in vivo has made it possible to exponentially increase knowledge of this layer in recent years. The latest development in OCT technology and the EDI system introduced by Spaide et al. [35] have provided increased penetration through the RPE, allowing for accurate deep choroidal imaging and measurements in vivo.

Several articles have shown the influence of the different methods used to control myopic progression on choroidal thickness. Regarding atropine, Yam et al. found a concentration-dependent response, with thicker choroids at higher atropine concentrations [36]. Concerning OK, Li et al. and Jin et al. concluded that OK treatment induced significant choroidal thickening [37,38]. Finally, some researchers have also studied the influences of the combination of several methods to control myopia on choroidal thickness. Chiang et al. showed additive effects of atropine and optical defocus at the level of the choroid [8], while Zao et al. found that the combination of OK and atropine induced a greater increase in SFChT than monotherapy with atropine [39].

It has been demonstrated that OK and soft peripheral-defocus-modifying CLs are effective in the control of myopic progression in children, retarding axial elongation. OK or corneal reshaping involves the use of specifically designed reverse geometry rigid GP lenses in overnight wear to reshape the cornea, inducing relative myopic shifts in peripheral refractive errors [40]. It has been postulated that choroidal thinning occurs early during myopic development and could be used as a clinical biomarker indicating myopia progression [41]; as such, several studies have shown the effect of OK on ChT in children. Gardner et al. showed that the choroid did not show long-term thickening during OK treatment in myopic children 11 to 15 years old [27]; however, other studies have demonstrated a thickening of the choroid associated with OK during short-term periods of wear. Loertscher et al. [23] showed a significant increase in ChT in children treated with OK in short follow-up periods (26 ± 18 days). Chen et al. [24] and Li et al. [25] showed a ChT increase after short-term OK treatment.

Three different types of MFSCL have been studied for myopia control in children: bifocal concentric lenses [17,20], peripheral gradient lenses [19], and extended depth of focus (EDOF) CLs [18]. MiSight CLs can be classified in the first group of bifocal or dual-focus lenses based on their concentric design with a concentric zone of rings with plus power addition for delivering simultaneously peripheral myopic defocus. There is only one manuscript describing the short-term effect (30 min) of monocular defocus imposed by two types of MCLs (Proclear multifocal ‘D’ Add +2.50 D and Proclear multifocal ‘N’ Add +2.50 D) on the ChT of 18 young myopic adults. The study showed that short periods of wearing different multifocal CL designs led to only small and nonsignificant changes in ChT [30].

Our study shows that ChT might be a predictor of the effectiveness of MiSight to control myopia. However, the sample size was small, which is a limitation of this study, as well as the high number of comparisons that were performed. It would be interesting to carry out a study with a larger sample size to confirm the results we have found.

## 5. Conclusions

There were no differences in the changes in choroidal thickness between children wearing MiSight CLs as the myopia control treatment and the control group wearing single-vision spectacle lenses after two years of treatment.

There were differences in the changes in choroidal thickness between responders and non-responders to MiSight. This could mean that choroidal thickness is a predictor of the effectiveness of MiSight in myopia treatment.

## Figures and Tables

**Figure 1 jcm-11-03833-f001:**
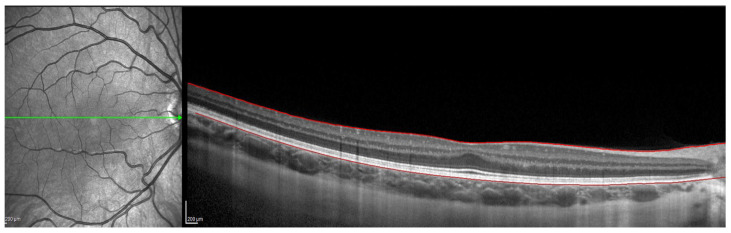
OCT image with lines in red corresponding to the retinal pigment epithelium and the internal limiting membrane.

**Figure 2 jcm-11-03833-f002:**
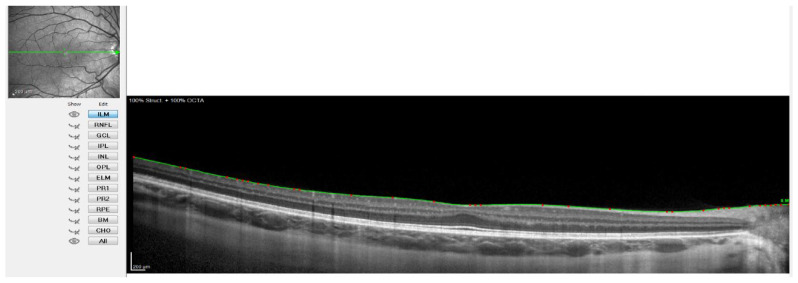
OCT image with the line of the internal limiting membrane marked in green, point by point.

**Figure 3 jcm-11-03833-f003:**
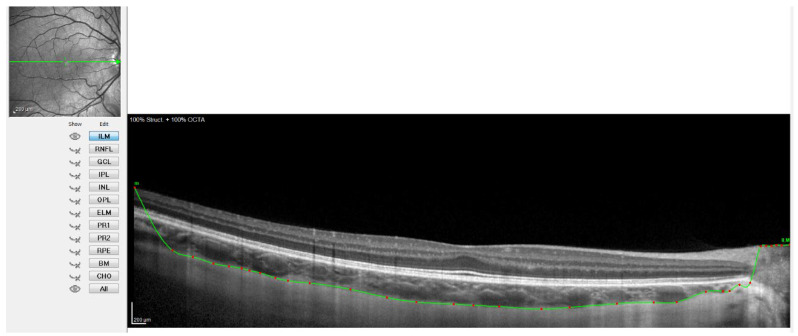
OCT image with the sclero-choroidal interface marked in green, point by point.

**Figure 4 jcm-11-03833-f004:**
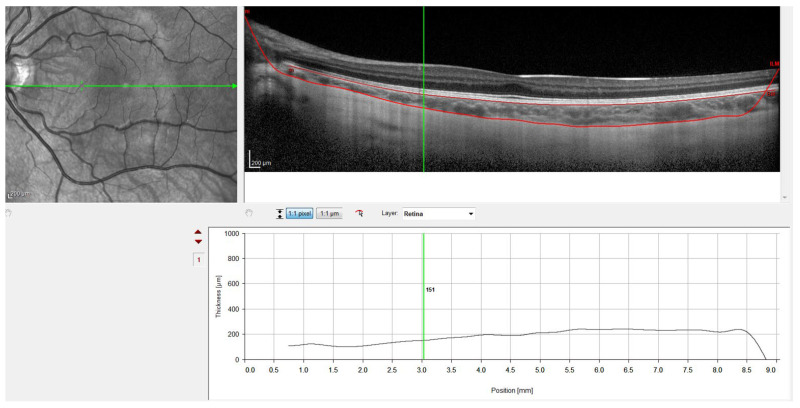
OCT image with lines in red corresponding to the retinal pigment epithelium and the sclero-choroidal interface.

**Table 1 jcm-11-03833-t001:** Changes in cycloplegic autorefraction (SE), axial length (AXL), mean keratometry, subfoveal choroidal thickness (SFChT), nasal choroidal thickness at 1 mm diameter (NChT 0.5 mm), temporal choroidal thickness at 1 mm diameter (TChT 0.5 mm), nasal choroidal thickness at 3 mm diameter (NChT 1.5 mm), and temporal choroidal thickness at 3 mm diameter (TChT 1.5 mm), at baseline and the 12- and 24-month follow-ups.

	MiSight (Mean ± SD)			SV (Mean ± SD)			*p* *
	Baseline	12 Months	24 Months	Baseline	12 Months	24 Months	
SE (D)	−2.16 ± 0.94	−2.34 ± 1.05	−2.61 ± 1.20	−1.75 ± 0.94	−2.18 ± 1.01	−2.48 ± 1.13	0.067
AXL (mm)	24.09 ± 0.55	24.21 ± 0.58	24.37 ± 0.59	24.00 ± 0.86	24.24 ± 0.86	24.45 ± 0.88	0.603
Keratometry (D)	44.24 ± 1.25	44.17 ± 1.20	44.21 ± 1.23	44.03 ± 1.59	44.06 ± 1.57	43.93 ± 1.62	0.533
SFChT (µ)	287.6 ± 53.45	285.1 ± 48.00	283.5 ± 43.38	283.6 ± 46.77	287.2 ± 43.22	296.7 ± 30.23	0.745
NChT 0.5 mm(µ)	280.1 ± 56.54	278.9 ± 49.03	274.7 ± 41.24	274.7 ± 44.99	278.6 ± 44.19	285.9 ± 31.15	0.667
TChT 0.5 mm(µ)	295.0 ± 54.30	288.2 ± 48.40	287.1 ± 44.32	289.0 ± 47.99	293.6 ± 43.80	300.8 ± 32.81	0.630
NChT 1.5 mm(µ)	257.2 ± 62.00	248.9 ± 57.49	242.7 ± 46.22	249.4 ± 52.73	247.4 ± 43.59	254.8 ± 36.05	0.570
TChT 1.5 mm(µ)	296.6 ± 54.86	286.6 ± 48.97	286.7 ± 43.93	291.5 ± 54.45	290.2 ± 50.09	303.0 ± 42.21	0.701

D: diopters; mm: millimeters; µ: microns. * *p*-value corresponds to the comparison between MiSight and SV groups’ variables at baseline.

**Table 2 jcm-11-03833-t002:** Changes (mean ± SD) in subfoveal choroidal thickness (SFChT), nasal choroidal thickness at 1 mm diameter (NChT 0.5 mm), temporal choroidal thickness at 1 mm diameter (TChT 0.5 mm), nasal choroidal thickness at 3 mm diameter (NChT 1.5 mm), and temporal choroidal thickness at 3 mm diameter (TChT 1.5 mm) between the first year of follow-up and baseline, and change in SFChT between the second year and the first year of follow-up for the MiSight and SV groups.

	∆ (12-Month Visit–Baseline)			∆ (24-Month Visit–12-Month Visit)		
	MiSight	SV	*p*	MiSight	SV	*p*
SFChT (µ)	0.472 ± 36.660	−1.966 ± 20.904	0.7375	−5.212 ± 24.697	0.600 ± 30.608	0.4870
NChT 0.5 mm(µ)	1.333 ± 39.555	−1.517 ± 26.203	0.7291	−5.424 ± 27.249	0.600 ± 35.022	0.5199
TChT 0.5 mm(µ)	−3.389 ± 37.112	−0.448 ± 22.214	0.6939	−5.333 ± 24.926	−4.267 ± 32.243	0.9009
NChT 1.5 mm(µ)	−6.500 ± 39.727	−6.966 ± 35.273	0.9608	−9.273 ± 35.952	8.267 ± 39.278	0.1347
TChT 1.5 mm(µ)	−10.33 ± 39.371	−6.483 ± 33.748	0.6779	−0.182 ± 31.062	0.200 ± 40.282	0.9715

µ: microns.

**Table 3 jcm-11-03833-t003:** Changes (mean ± SD) in subfoveal choroidal thickness (SFChT), nasal choroidal thickness at 1 mm (NChT 0.5 mm), temporal choroidal thickness at 1 mm (TChT 0.5 mm), nasal choroidal thickness at 3 mm (NChT 1.5 mm), and temporal choroidal thickness at 3 mm (TChT 1.5 mm) between the first year of follow-up and baseline, and change in SFChT between the second year and the first year of follow-up for responders and non-responders to the treatment with MiSight.

	∆(12-Month Visit–Baseline)			∆(24-Month Visit–12-Month Visit)		
	Responders	Non-Responders	*p*	Responders	Non-Responders	*p*
SFChT (µ)	17.200 ± 30.069	−11.48 ± 36.868	0.0183 *	−5.667 ± 30.677	−4.833 ± 19.306	0.9249
NChT 1 mm(µ)	16.533 ± 30.275	−9.524 ± 42.420	0.0498 *	−9.067 ± 34.872	−2.389 ± 19.358	0.5155
TChT 1 mm(µ)	14.467 ± 27.992	−16.14 ± 38.103	0.0124 *	−9.333 ± 26.567	−2.000 ± 23.714	0.4087
NChT 3 mm(µ)	6.867 ± 33.827	−16.05 ± 41.607	0.0880	−9.933 ± 41.687	−8.722 ± 31.643	0.9250
TChT 3 mm(µ)	9.933 ± 28.917	−24.81 ± 39.991	0.0071 *	−1.800 ± 33.435	1.167 ± 29.855	0.7896

µ: microns. * Significant values.

## Data Availability

The data presented in this study are available on request from the corresponding author. The data are not publicly available due to ethical issues.

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
