# Peer review of "Changes in the Choroidal Thickness of Children Wearing MiSight to Control Myopia"

_jcm, 2022, doi:10.3390/jcm11133833_

Round 1

Reviewer 1 Report

Two major drawbacks are noticeable in this study:

First, the method of image acquisition is not clear. How do the authors make sure that the measurements at the follow-up exams are taken from the same point as the first measurement? We know that there is high variability in choroidal thickness even in adjacent points, so it is important to be sure that we measure the same point in sequential measurements.

Second, statistical methods:

Sample size calculation: A standard deviation of 0.1 mm (100 microns) is too high for this measurement.

Sample size calculation: when you have a sample size calculation, you should consider it in the methods. A sample size of 8 patients in each group and then more than 30 patients. Why?

Too many calculations in finding differences in one variable. Between baseline and first follow up and again between first and second follow up. Too many comparisons increase the level of significance to a level much more than 0.05 which is noted in the study. This is not acceptable

Reviewer 2 Report

The authors described the importance of choroidal thickness in the development of myopia in this study which examined the effect of MySight contact lens. They found that choroidal thickness might work as predictor of the effectiveness of MySight in myopia treatment.

The manuscript is of interest for ophthalmologist and optometrist community and some issues must be solved prior to continue with the publication process.

1. Shorten the objective paragraph into one or two sentences

2. Divide the method into sections to improve the comprehension

3. In table 1 the p value difference should be interesting

4. Responders and no responders is a mixture of missight and SV? Detail and explain this issue
